# Temporal and Spatial Changes and Trend Predictions of Forest Carbon Sequestration Efficiency in China Based on the Carbon Neutrality Goal

Sixue Zhao [1], Wei Shi [1,*], Fuwei Qiao [2], Chengyuan Wang [3], Yi An [1] and Luyao Zhang [1]

1    College of Geography and Environmental Science, Northwest Normal University, Lanzhou 730030, China; 2021222978@nwnu.edu.cn (S.Z.); zhangluyao3367@163.com (L.Z.)
2    School of Economics, Northwest Normal University, Lanzhou 730070, China; qfw279@nwnu.edu.cn
3    Gansu Ecological Environment Science Design and Research Institute, Lanzhou 730022, China; wchy7514@163.com
*    Correspondence: shiwei0716@163.com; Tel.: +86-151-0132-8962

**Abstract:** Forestry's high-quality development is crucial for China's sustainable ecological, economic, and social progress. To elevate the efficiency of carbon sequestration in forestry, continuously improve the increment of carbon sinks, and contribute to achieving carbon neutrality, it is crucial to accurately assess the level of carbon sequestration efficiency in China's forestry and explore its long-term evolution trend. In this paper, a super-efficiency SBM model, which combines the SBM model with the super-efficiency method and considers the relaxation variables, was selected to evaluate the forestry carbon sequestration efficiency of 31 provinces in China; likewise, the temporal development features of the efficacy of Chinese forests in sequestering carbon were examined using the nuclear density estimation method. Secondly, the study constructed traditional and spatial Markov probability transfer matrices to further explore the spatiotemporal evolution of carbon sequestration efficiency within Chinese forestry. Finally, combined with the Markov chain infinite distribution matrix, the future trajectory of carbon sequestration efficiency in China's forestry was scientifically forecasted. The findings indicate that: (1) The average carbon sequestration efficiency of forestry in China showed a stable increase with fluctuations and reached the optimal state in 2018. The carbon sequestration efficiency level of various forest regions was always portrayed as southwest forest region > southern forest region > northeast forest region > northern forest region. From 2003 to 2018, there were significant differences in forestry carbon sequestration efficiency among provinces. The distribution of forestry carbon sequestration efficiency exhibited a "three-pillar" distribution pattern with Xizang, Zhejiang, and Heilongjiang as the core, and the marginal regions continuously promoted the carbon sequestration efficiency to the inland. (2) The type of transfer of forestry carbon sequestration efficiency in China is stable, and it is difficult to achieve cross-stage transfer in the short term. Moreover, the forestry carbon sequestration efficiency of each province tended to converge to a high (low) level over time, showing a "bimodal distribution" of low efficiency and high efficiency, indicating the existence of the obvious "club convergence phenomenon". (3) Forecasting from a long-term evolution trend perspective, the outlook for the future evolution of forestry carbon sequestration efficiency in China is optimistic, and the overall trend was concentrated in the high-value area. Therefore, future forestry development in China should contemplate both internal structure optimization and coordinated regional development. Attention should be placed on forestry carbon sequestration's role while considering the distinctive endowments of each region and developing reasonable, differentiated, and collaborative forestry management strategies.

**Keywords:** forestry; carbon sequestration efficiency; super efficiency SBM; spatial Markov chain model; trend forecasting

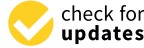



## 1. Introduction

For a long time, due to the increasing concentration of carbon dioxide caused by human activities, global warming has been a serious problem, which has aroused great concern in the international community. Global warming will not only pose a serious threat to the environment and ecosystem [1] but also have a significant impact on the growth of human society and economy. Therefore, the pursuit of net zero greenhouse gas emissions and green and low-carbon development has become the common responsibility of all countries in the world [2]. Although the international community has adopted several treaties and measures, like the Kyoto Protocol, the Paris Agreement, and the United Nations Framework Convention on Climate Change, to curb the increase in carbon emissions [3], the problem of increasing $CO_2$ emissions has still not been fundamentally solved [4] and still faces huge challenges [5]. Therefore, forest carbon sequestration, as the most economical and effective means of carbon sequestration, has been paid more attention to and favored in the process of realizing green and low-carbon development.

Forestry is a foundation for sustainable economic and social growth [6,7], pivotal in combating climate change and boosting regional production [8,9], and important in supporting the goal of carbon neutrality. The Paris Agreement defines carbon sinks and reservoirs, including forests [10], and calls on developing nations to enhance the role of these carbon sinks by protecting and effectively utilizing their forests. Thus, expanding forest carbon sequestration has been acknowledged globally as a crucial approach to combating climate change [11–13] and is pivotal in executing China's pledges to reduce carbon emissions. During the Paris United Nations Climate Conference, China made forestry development an essential project of its "nationally determined contribution". Moreover, it made a significant commitment to increase its forest reserves by 4.5 billion cubic meters compared to 2005 levels by 2030. Therefore, actively cultivating forest resources [14,15], strengthening forest carbon sequestration functions [16,17], and giving full play to the carbon offset role of forest carbon sinks [18–20] is the most practical and cost-efficient method of addressing worldwide warming [21,22].

Forest carbon sequestration has become a hot spot and research frontier for climate change mitigation. The United States and Germany are the leading countries in international research on forest carbon sequestration, and this study focused on multidisciplinary research on forest ecosystem carbon storage changes, biomass energy, and forest carbon sequestration capacity. Although China's research in forest carbon sequestration started relatively late, it has gained good momentum in recent years, and its international influence has also increased. More and more scholars focus on the cost of forest carbon sequestration, forest management, forestry policy formulation, and other fields, and have obtained specific research results on the mechanisms of forest carbon sequestration.

China's economy is going through a critical transition stage from rapid expansion to high-quality development [23]. As China's economy and society progress swiftly, its government consistently boosts its forestry investments, leading to emerging challenges [24]. For example, the proportions of mature forests and over-mature forests are increasing, and the land available for carbon sequestration is becoming more and more limited. The growth of the domestic forest coverage rate has gradually entered a bottleneck period, and it will not be sustainable to fulfill the function of forestry carbon sequestration alone by relying on the expansion of the area planted with trees. In addition, different regions have distinct forest resource endowments, and there are differences in afforestation costs and regional policies [25–28], which have led to severe problems of unbalanced and uncoordinated inter-regional forestry development. This urgently demands that China increase its emphasis on enhancing forest productivity and quality in its upcoming afforestation efforts, shifting from the scope and velocity type to the efficiency and quality type, from the emphasis on coverage to the change in storage, to achieve adequate growth of carbon sequestration capacity. Therefore, to reasonably evaluate and enhance the efficiency of forestry carbon sequestration [29] as a way to establish a new pattern of high-quality forestry advancement and accomplish regional carbon neutrality strategic goals, it is essential to maximize the

contribution of forestry carbon sequestration in the restricted forest area and strengthen regional coordination and complementarity [30,31].

Numerous academics have researched forestry carbon sequestration efficiency from various angles, as it is essential to the sustainable growth of forestry in China and can be used to estimate the value of forestry input and output. Xiong et al. [32] evaluated the forestry productivity of six provinces in Northwest China and identified six influencing factors. Chand et al. [33] employed stochastic frontier production analysis to estimate the productivity of local forestry in Nepal, which revealed that numerous socioeconomic and forest-related factors influenced the level of forestry efficiency. Through their research, Zang et al. [34] found that macro-policies, forestry science and technology input, allocation of resources, and regional GDP level impact the geographical distribution of forestry TFP in 30 Chinese provinces. Tian et al. [35] employed data enveloping analysis to determine the input–output efficiency of forestry in China from 1993 to 2010, and their findings revealed that the average comprehensive efficiency was 0.994. Xue et al. [36] studied the shifts and influencing elements of forest carbon sequestration efficiency in four primary forested areas of China, additionally assessing its efficiency convergence. Their findings indicated no σ convergence in the efficiency of forest carbon sequestration in China or the four major regions, but rather a distinct trend of absolute β divergence.

However, the studies on forestry carbon sequestration efficiency mentioned above primarily concentrated on a single efficiency measurement, spatial characteristics, and influences. There are a few investigations on the temporal and spatial evolution of forestry carbon sequestration efficiency and its long-term evolution trend. The efficiency evolution of forestry carbon sequestration, influenced via geographical conditions and economic growth, displays variations across time and space. Therefore, the spatial Markov chain examines the geographical closeness and correlation of efficiency changes in various regions while investigating forestry carbon sequestration efficiency's temporal and spatial evolution processes to predict and analyze efficiency changes scientifically.

Using the spatial panel data from the sixth period to the ninth period (2003–2018) of the continuous inventory of national forest resources in 31 provinces (municipalities and districts) of China, this study constructed a super-efficiency SBM model to estimate the carbon sequestration efficiency of Chinese forestry systematically and analyzed its spatiotemporal evolution characteristics. On this basis, by making the traditional and spatial Markov chain transfer probability matrix, we analyzed the trends in spatial and temporal development and the distribution patterns of forestry carbon sequestration efficiency in China. Additionally, we projected its long-term evolutionary trend. The impact of neighboring areas on forestry carbon sequestration efficiency in the long-term evolution process was intensely discussed, providing suggestions for China to deal with worldwide warming, enhance forest carbon sink capacity, and formulate differentiated forestry development policies [37].

## 2. Data and Methods

### 2.1. Data Source and Index System Construction

This work aimed to calculate the carbon sequestration efficiency of Chinese forestry from an input–output viewpoint. The research object covers 31 provinces in China (data from Macao, Taiwan, and Hong Kong are temporarily lacking), and the study spans from 2003 to 2018. Since 2003 is the conclusion year of China's sixth national forest resources inventory, 2003 was chosen as the starting year. Given data availability, 2018, the end year of the ninth nationwide forest inventory, was selected as the end year. According to the regional division and geographical distribution characteristics of forestry in the China Forestry Development Report, China's 31 provinces were categorized into four distinct forest regions, including the northeast forest region (Inner Mongolia, Liaoning, Jilin, and Heilongjiang); the southern forest region (Shanghai, Jiangsu, Zhejiang, Anhui, Fujian, Jiangxi, Hubei, Hunan, Guangdong, Guangxi, Hainan, and Guizhou); the southwest forest region (Yunnan, Tibet, Sichuan, and Chongqing); and the northern forest region

(Beijing, Tianjin, Hebei, Shanxi, Shandong, Henan, Shaanxi, Gansu, Qinghai, Ningxia, and Xinjiang) [38]. In view of the limitations and availability of statistical data, and taking into account the problem of consistency and comprehensive continuity of statistical data, the original data used in this study to construct the input–output index system (Table 1) were all from the fourth to the ninth National Forest Resources Inventory of China, *China Forestry Statistical Yearbook* (2003–2018), and *China Statistical Yearbook* (2003–2018), and other data were found on the official websites of the National Bureau of Statistics and the State Forestry Administration.

**Table 1.** Index system of forestry carbon sequestration efficiency in China.

| Index | Secondary Index | Three-Level Index | Unit |
|---|---|---|---|
| Input index | Labor force | Forestry system employees at the end of the year | / |
| | Capital | Investment in fixed assets in forestry completed | / |
| | Land | Afforestation area | hectare |
| Output indicator | Expected output | Forestry primary sector output | / |
| | | Value of carbon sequestration in forestry | / |
| | | Forest greening rate | Percentage (%) |

Forestry factors are chosen from land, labor, and capital [39]. (1) The completion of forestry fixed asset investment, the total number of employees in the forestry system at the end of the year, and the area planted with trees are the three main selection criteria for forestry input indicators. The finished amount of forestry fixed value investment, among them, primarily indicates the investment scale of forestry construction funds, which reflects the size of capital investment. Employees in the forestry system at the end of the year denote the labor input index, which measures the amount of labor input in forestry construction each year. The land input index selected the afforestation area. (2) Regarding forestry output indicators, forestry output indicators mainly include three points: Forestry has ecological, economic, and social benefits [40]. The output value of the primary industry generally expresses forestry economic benefits. However, as an emerging ecological service product, forestry carbon sequestration is essentially a market-based compensation and cash mechanism for ecological benefits, and it will also produce certain economic benefits in the process of carbon market trading [41], which should be included in China's forestry economic accounting system. Therefore, to quantify the economic value of forestry carbon sequestration in China, this paper adopted the forest stock expansion method to calculate forest carbon sequestration (the forest stock expansion method is used to convert the stock into cost-effective carbon sequestration according to the different carbon sequestration capacities of different regions and different tree species using the conversion coefficient. Forest carbon sequestration refers to the carbon elements fixed in the soil organic carbon pool in the forest ecosystem at a certain point in time, such as underground and above-ground biomass, litter and dead wood, etc., which is the result of the accumulation of forest ecosystems for many years). Secondly, this paper selected "10 USD/t carbon" as the reference price from the Comparison Table of the World Carbon Sink Trade Price Forecast compiled by Dr. Lin Derong [42], converting the carbon sequestration price according to the average exchange rate of the current year. Finally, the value of carbon sequestration in Chinese forestry was obtained by multiplying the carbon sink price with the amount of carbon sequestration in forestry (due to space limitations, the calculation process for the amount and value of carbon sequestration in forestry was not reflected in this paper). To summarize, "the output value of the primary industry of forestry" and "the value of forestry carbon sink" were used to illustrate the economic benefits of forestry in this article.

The term "forest greening rate" refers to the proportions of shrub, forest, farming, and surrounding tree cover areas of the total land area; this rate was used in this study to illustrate the ecological benefits of forestry [43]. Due to the difficulty in measuring forestry social benefits, this paper did not consider forestry social benefits in the forestry output index [44].

Since the efficiency measured via the DEA method is only a relative efficiency, the six indicators selected in this paper, although they cannot fully cover the input–output factors, basically consider the main factors of forestry input and output without considering the social benefit index of forestry, and the measurement results are still representative and can truly reflect the carbon sequestration efficiency of forestry.

*2.2. Research Methods*

2.2.1. The Super-Efficiency SBM Model

After the DEA method is improved twice by Tone, the ultra-efficient SBM model can further analyze the DMU with an efficiency value of 1 based on the existing SBM model. [45]. The super-efficiency SBM model is a method to evaluate and optimize the efficiency of a system. Based on the data enveloping analysis model, it can calculate the efficiency of the system more accurately by quantifying and weighting the input and output indexes of the system and considering the relaxation variables. Compared with the previous ultra-efficient methods, the super-efficiency SBM model has the advantages of more flexibility and accuracy. As a result, this study investigated the issue using the super-efficiency SBM model. $X$ and $Y$ stand for the input and output variables; $y^g$ and $y^b$ denote the expected and undesirable results; and $m$ and $s$ stand for the number of input and output variables, respectively ($s_1$ and $s_2$ denote the desired and undesirable results, respectively, i.e., ($s_1 + s_2 = s$)). Matrix $X$, $Y^g$, and $Y^b$ can be stated as $X = (x_{i,j}) \in R^{m \times n}$, $Y^g = \left( y_{i,j}^g \right) \in R^{s_1 \times n}$, and $Y^b = \left( y_{i,j}^b \right) \in R^{s_2 \times n}$, respectively; $X$, $Y^g$, and $Y^b$ were set to greater than 0.

Additionally, we defined the set of production possibilities as: $P = \{(x, y^g, y^b) | x \geq X\lambda,$ $y^g \geq Y^g\lambda, y^b \leq Y^b\lambda, \lambda \geq 0\}$. While $X$, $Y^g$, and $Y^b$ represent the input, expected output, and unexpected output matrices, respectively, the model is defined as follows:

$$\rho * = \min \frac{\frac{1}{m} \sum_{i=1}^{m} \frac{\overline{x_i}}{x_{ik}}}{\frac{1}{s_1 + s_2} \left( \sum_{r=1}^{s_1} \frac{\overline{y_r^g}}{y_{rk}^g} + \sum_{r=1}^{s_2} \frac{\overline{y_l^b}}{y_{lk}^b} \right)} \tag{1}$$

$$\text{s.t.} \begin{cases} \overline{x} \geq \sum_{j=1, \neq 0}^{n} \lambda_j x_i \\ \overline{y}^g \leq \sum_{j=1, \neq 0}^{n} \lambda_j y_j^g \\ -b \geq \sum_{j=1, \neq 0}^{n} \lambda_j y_j^b \\ \overline{x} \geq x_k, \overline{y}^g \leq y_k^g, y^b \geq y_k^b, \overline{y}^g \geq 0, \overline{y}^b \geq 0, \lambda \geq 0 \end{cases} \tag{2}$$

where $\rho *$ represents forestry carbon sequestration efficiency; $n$ indicates the number of decision units; and $x_{ij}$, $y_{rj}^g$, and $y_{lj}^b$ represent the input, expected output, and non-expected output of the evaluated unit, respectively. $\lambda$ represents the weight vector; $m$, $s_1$, and $s_2$ denote the index numbers of DMU's input variables, expected output variables, and non-expected output variables, respectively; the relaxation variables of input factors, expected output factors, and non-expected output factors are represented by $\overline{x}$, $\overline{y}_r^g$, and $\overline{y}_l^b$, respectively; in the improved DMU, $x_{ik}$, $y_{rk}^g$, and $y_{lk}^b$ represent the ideal input amount, expected output, and unexpected output, respectively.

2.2.2. Kernel Density Estimation

The probability density function has been mostly evaluated via kernel density estimation, and continuous density curves illustrate the distribution and evolution characteristics

of forestry carbon sequestration efficiency. Formulas (3) and (4) outline the equation of kernel density estimation [46]:

$$f(x) = \frac{1}{Nh}\sum_{i=1}^{N} K\left(\frac{X_i - \overline{X}}{h}\right) \tag{3}$$

$$K(x) = \frac{1}{\sqrt{2\pi}}e^{\frac{-x^2}{2}} \tag{4}$$

where $f(x)$ represents the density function; $N$ represents the number of samples observed; $K(\cdot)$ indicates the kernel function; $X_i$ denotes the specific observation; $\overline{X}$ denotes the mean; and h stands for bandwidth.

### 2.2.3. The Spatial Markov Chain Model

By creating the probability matrix of state transition, which may reflect the regional state and its downward or upward mobility, the Markov chain model can identify the state of event occurrence and its changing trend [47]. The classic Markov chain model was employed in this study to build an N × N Markov transfer matrix of probability according to forestry carbon sequestration efficiency (Table 2). Using the quartile division standard, the forestry carbon sequestration efficiency of 31 provinces was divided into four states: low efficiency, relatively low efficiency, relatively high efficiency, and high efficiency, denoted via k = 1, 2, 3, and 4, respectively. The transfer from low to high was described as upward, while the transfer from high to low was described as downward, so as to explore the dynamic evolution characteristics of efficiency. The transfer probability of forest carbon sequestration efficiency in a province from t-year state $E_i$ to t + 1-year state $E_j$ is as follows:

$$P_{ij}(E_i \rightarrow E_j) = \frac{n_{ij}}{n_i} \tag{5}$$

where $P_{ij}$ represents the transfer probability of forest carbon sequestration efficiency in a province from t-year state $E_i$ to t + 1-year state $E_j$, and the transfer frequency may indicate the efficiency transfer probability. $n_{ij}$ represents the total number of areas from tiers $i$ to $j$. When forestry carbon sequestration efficiency shifts from state $E_i$ to state $E_j$, $n_i$ denotes the number of provinces whose state $E_i$ is at level $i$.

**Table 2.** The Markov chain state transition probability matrix.

| t/(t + 1) | 1 | 2 | 3 | 4 |
|:---:|:---:|:---:|:---:|:---:|
| 1 | $P_{11}$ | $P_{12}$ | $P_{13}$ | $P_{14}$ |
| 2 | $P_{21}$ | $P_{22}$ | $P_{23}$ | $P_{24}$ |
| 3 | $P_{31}$ | $P_{32}$ | $P_{33}$ | $P_{34}$ |
| 4 | $P_{41}$ | $P_{42}$ | $P_{43}$ | $P_{44}$ |

N = 4; t represents the initial year, and t + 1 represents the year following the initial year.

The "spatial lag" was added to the Markov probability transfer matrix based on the standard Markov chain model, which fixes the problem with the traditional Markov chain neglecting spatiality [48]. The N × N Markov chain transition probability matrix could be divided into an N × N × N transition probability matrix by adjusting the spatial weight matrix (Table 3). To demonstrate the impact of geographical factors on forest carbon sequestration efficiency, $P_{ij}(N)$ indicates the transition probability of forest carbon sequestration efficiency from t-year state $E_i$ to t + 1-year state $E_j$ when the lag type of the assessment unit is $N_i$. The formal expression is as follows:

$$Lag = Y_i W_{ij} \tag{6}$$

where *Lag* denotes the spatial lag value of the evaluation unit *I*; $Y_i$ represents the attribute value of the evaluation unit *I*; and $W_{ij}$ is a matrix of interactions between evaluation units and neighboring regions.

**Table 3.** Conditional probability matrix of state transition of the spatial Markov chain.

| Spatial Lag | t/(t + 1) | 1 | 2 | 3 | 4 |
|---|---|---|---|---|---|
| 1 | 1 | $P_{11/1}$ | $P_{12/1}$ | $P_{13/1}$ | $P_{14/1}$ |
| | 2 | $P_{21/1}$ | $P_{22/1}$ | $P_{23/1}$ | $P_{24/1}$ |
| | 3 | $P_{31/1}$ | $P_{32/1}$ | $P_{33/1}$ | $P_{34/1}$ |
| | 4 | $P_{41/1}$ | $P_{42/1}$ | $P_{43/1}$ | $P_{44/1}$ |
| 2 | 1 | $P_{11/2}$ | $P_{12/2}$ | $P_{13/2}$ | $P_{14/2}$ |
| | 2 | $P_{21/2}$ | $P_{22/2}$ | $P_{23/2}$ | $P_{24/2}$ |
| | 3 | $P_{31/2}$ | $P_{32/2}$ | $P_{33/2}$ | $P_{34/2}$ |
| | 4 | $P_{41/2}$ | $P_{42/2}$ | $P_{43/2}$ | $P_{44/2}$ |
| 3 | 1 | $P_{11/3}$ | $P_{12/3}$ | $P_{13/3}$ | $P_{14/3}$ |
| | 2 | $P_{21/3}$ | $P_{22/3}$ | $P_{23/3}$ | $P_{24/3}$ |
| | 3 | $P_{31/3}$ | $P_{32/3}$ | $P_{33/3}$ | $P_{34/3}$ |
| | 4 | $P_{41/3}$ | $P_{42/3}$ | $P_{43/3}$ | $P_{44/3}$ |
| 4 | 1 | $P_{11/4}$ | $P_{12/4}$ | $P_{13/4}$ | $P_{14/4}$ |
| | 2 | $P_{21/4}$ | $P_{22/4}$ | $P_{23/4}$ | $P_{24/4}$ |
| | 3 | $P_{31/4}$ | $P_{32/4}$ | $P_{33/4}$ | $P_{34/4}$ |
| | 4 | $P_{41/4}$ | $P_{42/4}$ | $P_{43/4}$ | $P_{44/4}$ |

N = 4; t represents the initial year, and t + 1 represents the year following the initial year.

The stationary distribution of a random process could be found by computing the limits of the Markov transfer probability matrix, and the dynamic development trend of forest carbon sequestration efficiency can be anticipated. By the definition of limits:

$$\lim_{k\to\infty} \omega(k) = \lim_{k\to\infty} \omega(k+1) = \omega \tag{7}$$

By bringing Formula (7) into the recursive formula of the Markov prediction model:

$$\lim_{k\to\infty} \omega(k+1) = \lim_{k\to\infty} \omega(k)P \tag{8}$$

where $\omega$ represents the ultimate state matrix of Markov process evolution; the conditions satisfied by the final state matrix are $\omega = \omega P$, $0 \le \omega i \le 1$, and $\sum_{i=1}^{n} \omega_i = 1$. The conventional Markov chain is a method for estimating the ultimate state of a spatial Markov process. According to this principle, the final state of the Markov process is computed under various spatial hysteresis conditions, and the effect of spatial hysteresis on the evolution trend of the Markov process can be projected.

## 3. Results

### 3.1. Estimation and Time Series Analysis of Forestry Carbon Sequestration Efficiency

The super-efficiency SBM model was applied in this article to assess the forestry carbon sequestration efficiency of 31 Chinese provinces from 2003 to 2018, and the mean efficiency of different forest regions was compared and analyzed (Figure 1). The results are as follows:

(1)　As shown in Figure 1, China's forestry carbon sequestration efficiency increased year by year from 2003 to 2018, reaching the highest level in history in 2018. This was mainly because China has formulated more scientific forestry policies in recent years. For example, the state has improved the top-level design of forest ecosystem management; regional carbon sequestration targets are set according to local conditions. To effectively improve the efficiency of forestry management, a scientific carbon sink accounting method system has been established. The construction of a carbon trading market and state investment should be simultaneously carried out to effectively pro-

mote the continuous improvement of forestry carbon sequestration efficiency. The creation of a forestry policy is more scientific, which supports the ongoing enhancement of forestry production efficiency. Regarding development trends, the efficiency of forestry carbon sinks follows the pattern of southwest forest region > southern forest area > northeast forest region > northern forest region. Furthermore, the four forest regions' input–output efficiency values significantly improved between 2010 and 2012. During this time, the State Forestry Administration announced the Forestry Industry Revitalization Plan (2010–2012), which provided some policy support to help optimize the structure of the forestry industry, ensure China's forestry's sustainable and healthy development, and comprehensively and effectively improve forestry efficiency.

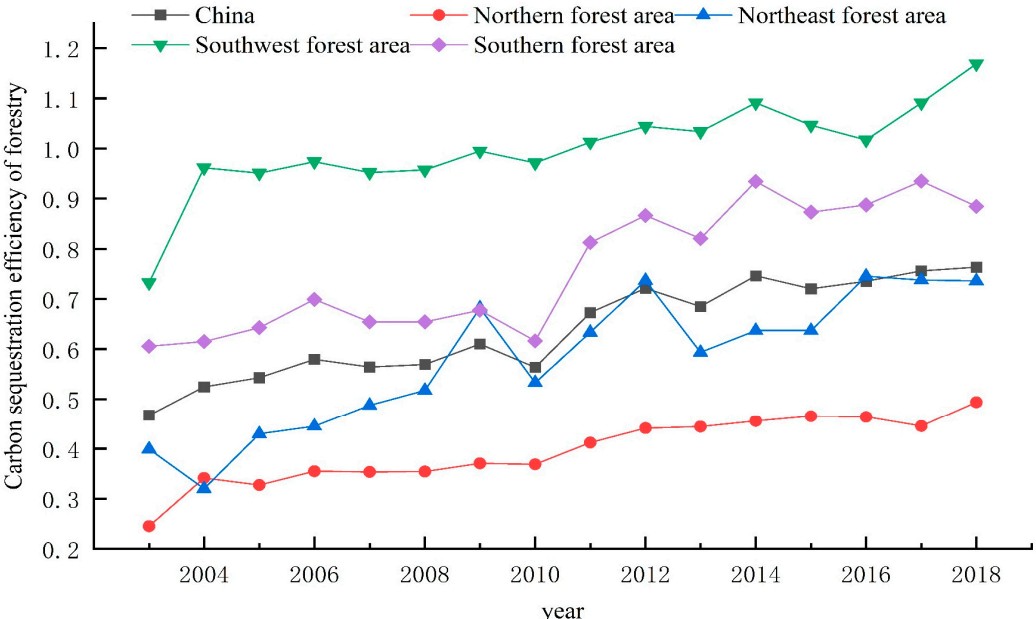

**Figure 1.** Trend of forestry carbon sequestration efficiency from 2003 to 2018.

(1)　　Kernel density of forestry carbon sequestration efficiency in China.

(2)　From 2004 to 2016, the southwest forest region's average forestry efficiency remained around one, always ahead of the national and other forest areas' averages. In 2003, the efficiency of the southwest forest region was the lowest, at only 0.733, while in 2018, it was the highest, at 1.169. This indicates that the efficiency of forest carbon sequestration in the southwest forest area is constantly improving. During this time, the southwest forest region effectively transformed the factors' input and output with excellent quality, consistent with the actual condition. The southwest forest region, China's second-largest natural forest region, depends on rich forest resource advantages and outstanding forestry management technology to effectively achieve the rational allocation of forestry resources and offer significant carbon sequestration benefits to China.

(3)　The southern forest region's average efficiency stayed in second place for an extended period, and its efficiency value sank to the bottom in 2010 but quickly rose in 2011, reached its peak in 2014, and then exhibited a continuous fall, but the decline was moderate. Overall, the southern forest region's efficiency value demonstrated a positive pattern of continual increase.

(4)　With an average of 0.58, the change in efficiency of the northeast forest region is typically comparable to that of the entire nation. From 2008 to 2013, the efficiency value of the northeast forest area fluctuated wildly, showing an inverted "W" shape and slowly rising, and the forestry efficiency was greatly improved. It began to level

off in 2016. The reason for its improvement could be that the state started a fresh round of the "Three North" shelterbelt project, which considerably elevated the forestry ecological value of the northeast forest region [49,50].

(5)  With the overall forestry efficiency in the northern forest area being low and there still being room for improvement, the average efficiency value of the northern forest area is lower than the national average. It has been at the bottom for a long time. The northern forest region's low efficiency may be due to its relative lack of forest resources, unfavorable climatic conditions, and somewhat outdated forestry technologies compared to the other three forest areas.

The observation years from 2003 to 2018 were employed in this study to further investigate the evolution of forest carbon sequestration efficiency over time in China and four forest areas. The kernel density estimation method described the distribution location, status, and scalability of each forest region's efficiency. The evolutionary traits of forest carbon sequestration efficiency through time were finally summed together.

Figure 2 illustrates China's kernel density map of forestry carbon sequestration efficiency. The distribution position indicates that the efficiency of carbon sequestration in China's forests is steadily rising, with the center of efficiency consistently shifting to the right. The kernel density function curve during the sample study period displayed a bimodal distribution from the standpoint of distribution pattern, demonstrating that China's forestry efficiency as a whole has a substantial polarization characteristic. Additionally, the peak value of the kernel density function's central peak showed a fluctuation decline, while the peak's shape gradually relaxed, and the peak value of the lateral height showed a trend of increasing year over year, indicating that the aggregation degree of China's low-forestry-efficiency provinces was steadily declining while the aggregation degree of areas with high efficiency was steadily rising. The right tailing of the kernel density function of forestry efficiency in China steadily diminished from the standpoint of distribution ductility, suggesting a steady reduction in the spatial variation of forestry efficiency across the nation.

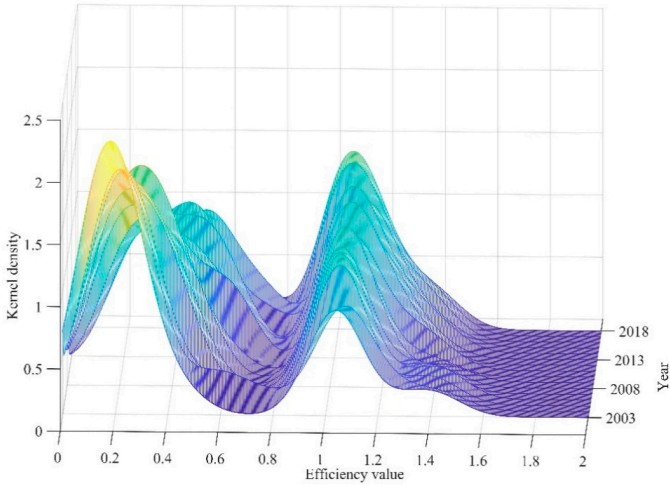

**Figure 2.** Distribution dynamic change in forestry efficiency in China.

(2)  Kernel density of forestry carbon sequestration efficiency in four forest regions.

Figure 3 depicts the kernel density map of forestry carbon sequestration efficiency in the northern forest region. The center of forestry efficiency in the northern forest region moved first to the right and then to the left, indicating that the carbon sequestration efficiency first increased and then decreased. From the perspective of distribution pattern, the kernel density function curve during the whole sample study period presents an apparent distributed bimodal distribution. Moreover, from the standpoint of distribution pattern, the kernel density function curves of the entire sample study period showed a

precise bimodal distribution, indicating that the carbon sequestration efficiency of different provinces in the northern forest region was vastly different. In addition, the edge peak-to-peak value of the kernel density function increased significantly in 2018, and the peak shape decreased steadily, indicating that the concentration of provinces with better efficiency values was growing and that the inter-provincial gap was gradually narrowing.

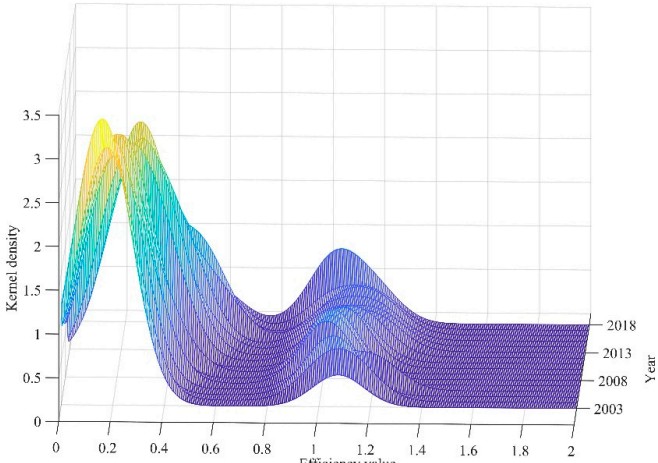

**Figure 3.** Distribution dynamic changes in efficiency growth in the northern forest area.

Figure 4 shows the core density map of forestry carbon sequestration efficiency in the northeast forestry region. From the distribution position, the efficiency center moved roughly to the right, demonstrating that the carbon sequestration efficiency level in the northeast forest area appears to be rising. From the distribution pattern, the curve of the kernel density function exhibits an obvious bimodal distribution, indicating that the carbon sequestration efficiency of this forest area has a significant polarization feature on the whole. The peak value of the kernel density function's side wave significantly increased during the sample investigation period. This indicates that the northeast forest region's concentration of provinces with high carbon sequestration efficiency grew and that the efficiency gap was gradually closing.

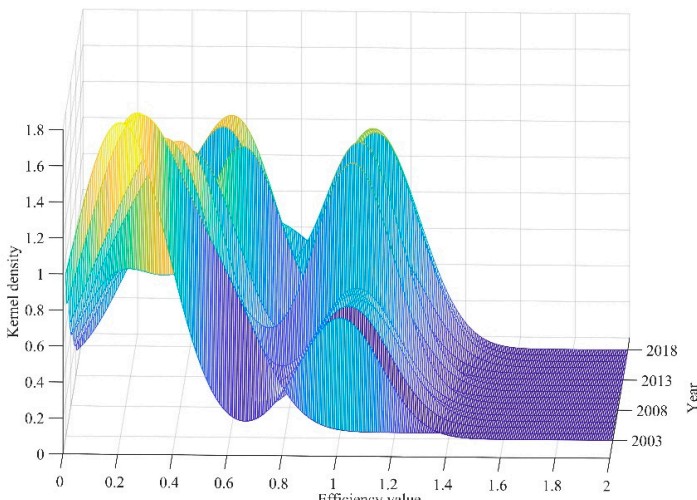

**Figure 4.** Distribution dynamic changes in efficiency growth in the northeast forest area.

Figure 5 depicts the kernel density map of forestry carbon sequestration efficiency in the southwest forest region. The efficiency center moved to the right, indicating improved efficiency. From the perspective of distribution pattern, the kernel density distribution curve changed from the "three-peak" distribution pattern in 2003 to the "two-peak" distribution

pattern, indicating that the forest carbon sequestration efficiency changed from the multi-polar differentiation characteristic to the significant polarization characteristic. The second wave peak value was significantly higher than the first, indicating that the provinces with high carbon sequestration efficiency in the southwest forest area have a higher degree of agglomeration. During the sample investigation, the lateral peak value of the kernel density function continued to rise, and the wave crest shape gradually narrowed, indicating that the concentration degree of provinces with high carbon sequestration efficiency in the southwest forest region continued to grow and the gap in carbon sequestration efficiency gradually narrowed.

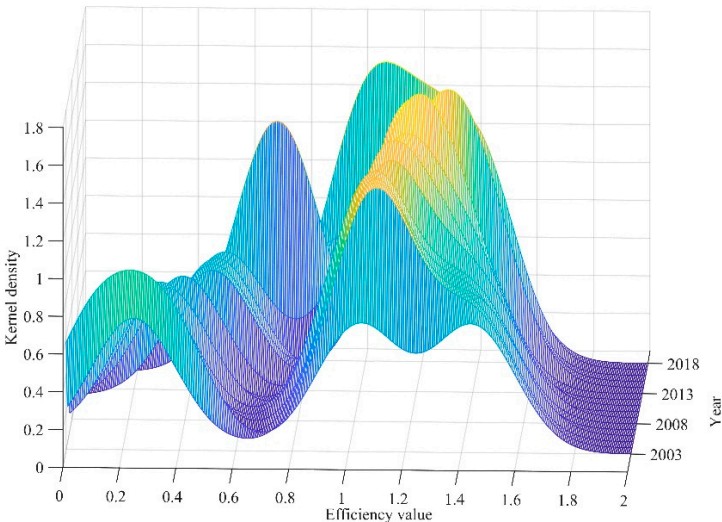

**Figure 5.** Distribution dynamic changes in efficiency growth in the southwest forest area.

Figure 6 illustrates the kernel density map of forestry carbon sequestration efficiency in the southern forest region. The efficiency center moved from the distribution position to the right, indicating that the forestry efficiency in the southern forest region was constantly improving. The kernel density function distribution pattern exhibited an apparent bimodal distribution, indicating that forestry efficiency in the southern forest region had strong polarization characteristics. During the sample investigation, the lateral peak value of the kernel density function significantly increased, and the wave crest shape gradually narrowed from a broad peak to a sharp rise, indicating that the aggregation degree of provinces with high forestry efficiency continued to increase and the inter-provincial difference degree gradually weakened.

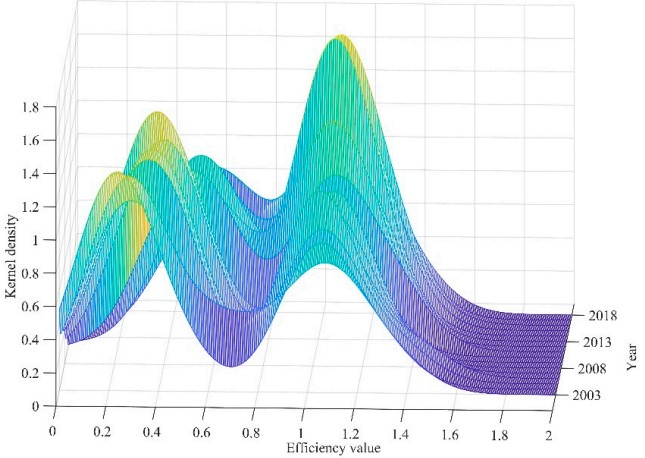

**Figure 6.** Distribution dynamic changes in efficiency growth in the southern forest area.

### 3.2. Temporal and Spatial Change Characteristics of Forestry Carbon Sequestration Efficiency

3.2.1. Temporal Change Characteristics of Forestry Carbon Sequestration Efficiency

Time series analysis, kernel density estimation, and spatiotemporal pattern analysis were employed to determine the change trend and spatial evolution difference of forest carbon sequestration efficiency over time to understand the spatiotemporal patterns of this efficiency. A conventional Markov chain transition probability matrix (Table 4) and a spatial Markov transition probability matrix were built to further examine its spatiotemporal evolution properties.

**Table 4.** Markov transfer probability matrix of forestry efficiency in China from 2003 to 2018.

| t\t + 1 | 1 | 2 | 3 | 4 | n |
|---|---|---|---|---|---|
| 1 | 0.8083 | 0.1750 | 0.0083 | 0.0083 | 120 |
| 2 | 0.0847 | 0.7373 | 0.1695 | 0.0085 | 118 |
| 3 | 0.0000 | 0.0804 | 0.7321 | 0.1875 | 112 |
| 4 | 0.0000 | 0.0174 | 0.1478 | 0.8348 | 115 |

Table 4 outlines a traditional Markov transfer probability matrix of China's forestry carbon sequestration efficiency from 2003 to 2018. According to the calculation findings, it is possible to obtain: (1) In terms of maintaining state stability, the primary diagonal value of the Markov chain transfer probability matrix was significantly higher than the non-diagonal value, indicating that the type transfer of carbon sequestration efficiency in Chinese forestry is stable and the probability of maintaining the original state is significant. (2) In terms of extreme convergence, the values of the two ends of the diagonal value (type 1 and type 4) were higher than those of the middle (type 2 and type 3), indicating that the forestry efficiency tends to converge to an increased (low) level. A "club convergence" phenomenon was present. Under the assumption that the difference in efficiency between high- and low-efficiency areas has not shrunk, this phenomenon primarily indicates that the internal difference in efficiency between high- and low-efficiency areas has gradually shrunk, resulting in a continuous weakening of the regional disparities in forestry carbon sequestration efficiency. (3) In terms of state transition, the maximum probability on the non-diagonal line (0.1875) was significantly lower than the minimum probability on the diagonal line (0.7321), indicating that achieving cross-stage transition in the short term is challenging. In terms of state transition prediction, the possibility of type 2 moving to type 1 was smaller than the probability of type 2 moving to type 3, while the probability of type 3 moving to type 2 was smaller than the probability of type 3 moving to type 4, namely $P_{21}(0.0847) < P_{23}(0.1695)$, $P_{32}(0.0804) < P_{34}(0.1875)$, indicating that the likelihood of forestry carbon sequestration efficiency moving to a low state is relatively low.

3.2.2. Spatial Distribution Characteristics of Forestry Carbon Sequestration Efficiency

In order to comprehend the geographical evolution features of forestry carbon sequestration efficiency in China, this article drew a spatial distribution map of forestry carbon sequestration efficiency in 31 provinces using ArcGIS 10.8 software (Esri, Redlands, CA, USA) (Figure 7). As shown in Figure 7, the efficiency of forestry carbon sequestration in China increased dramatically between 2008 and 2013. Except for Xinjiang, which was always in the low-value area (0~0.2), the other provinces showed varying degrees of increase by 2013. This was mainly owing to the country's significant support for land greening and the growth of the forestry industry. The year 2013 was critical for completing the tasks of the 12th Five-Year Plan. China is dedicated to attaining "double growth" in forestry and growing the carbon sink in forestry, and it actively engages in carbon sequestration afforestation. The extent of carbon sequestration afforestation hit a new high in 2013, and tremendous results were realized. Significant improvements in forestry efficiency may be found mainly in the northeast and southern forest regions. These two areas offer rich forest resources, ideal forest growth circumstances, and a broad range of trees, and have

traditionally been vital locations for China's forestry development. Since the year 2003, China has improved its forest resource management, and the capacity and efficiency of the two forest regions for carbon sequestration have been considerably enhanced due to suitable factor input and allocation. For example, Heilongjiang is one of the first provinces in China to start carbon sequestration afforestation and develop the forestry economy. Since 2003, it has implemented natural forest protection, built farmland shelterbelts, established ecological demonstration zones and nature reserves, and included the forestry industry in the ten key industrial projects in 2011. Inner Mongolia is Northern China's most vital environmental defensive line, and its ecological position is notable. With the execution of the western growth strategy, Inner Mongolian forestry has reached a rapid growth stage and achieved exceptional results. In 2013, Jilin Province fiercely executed the green transformation development plan in forestry, emphasizing the growth of the forest tourist sector and driving the overall development of the forestry industry in a market-oriented manner, and strived to realize the leap from a province with significant forest resources to a province with a strong forestry economy. During the study period, China actively promoted the development of coastal shelterbelts and Yangtze River basin shelterbelts, emphasizing the increase in forest vegetation, the construction of mangroves and coastal dry forest belts, vigorously developing the characteristic forestry in coastal areas, and the structure of a complete shelterbelt system in the Yangtze River basin. The abovementioned policy could explain the two forest regions' continual increase in efficiency.

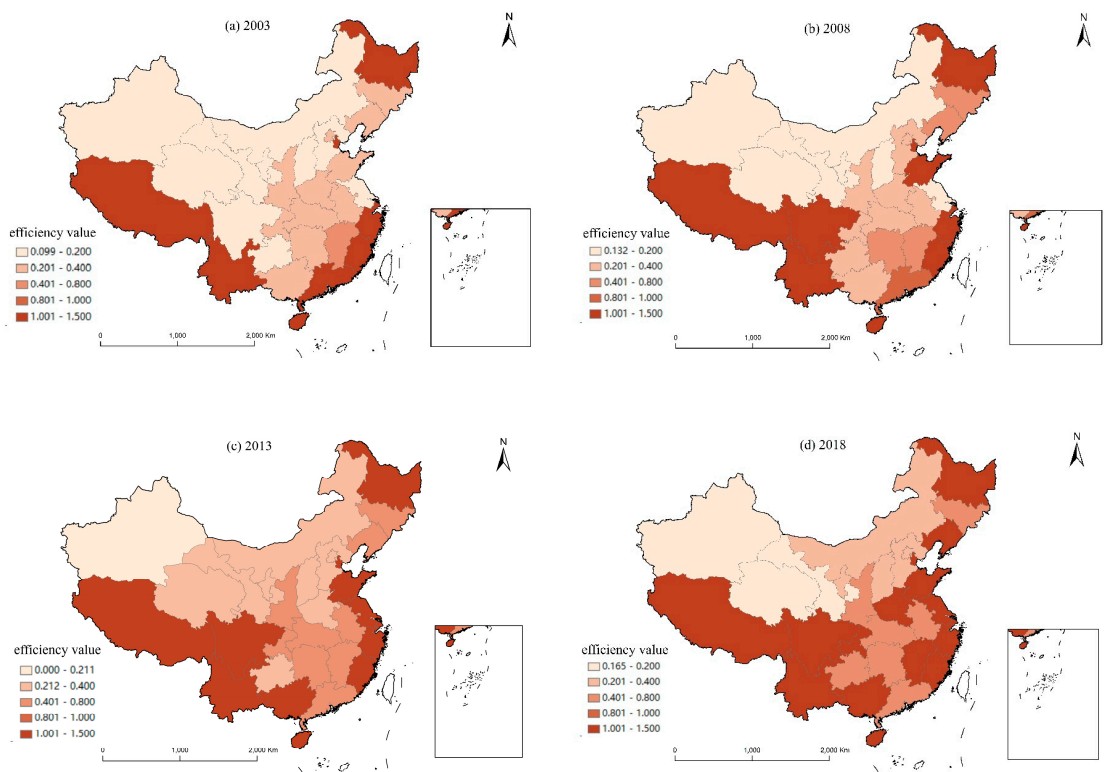

**Figure 7.** Distribution of forestry carbon sequestration efficiency in China from the years 2003–2018 (**a–d**).

Chongqing, Sichuan, Yunnan, and Tibet all belong to the southwest forest area, among which the forestry efficiencies of Sichuan, Yunnan, and Tibet have always maintained a high efficiency level above 1.000 during the study period and reached the optimal state of input/output. By the year 2018, Chongqing had also joined the ranks of high-efficiency levels. For a long time, most provinces in the northern forest region have had poor efficiency levels. The main reasons are low forest coverage, a lack of forestry resource endowment, complex site conditions, and a sparse population in the northern forest area, all of which impede economic

development, lead to relatively backward forestry innovation technology, and limit high-quality forestry development. This could be the primary explanation for the northern forest's low carbon sequestration efficiency. With the continuous implementation and development of crucial shelterbelt system construction projects, such as the Three North Areas [51] and the Yangtze River Basin [52], key projects, such as the Beijing–Tianjin sandstorm source control project [53,54], and the continuous updating and improvement of forestry policies, forestry efficiency in the northern forest region will be significantly improved.

In summary, from 2003 to 2018, China's forestry carbon sequestration efficiency was generally low and needed to be further improved. In terms of efficiency distribution, Xizang, Yunnan, Zhejiang, Fujian, and Heilongjiang are mainly high-efficiency gathering areas, and the overall trend of "three pillars" surrounds them, driving the efficiency level of forestry carbon sequestration gradually to the inland.

The spatial Markov transfer probability matrix was created in this paper to compare and analyze the transfer probability of forest carbon sequestration efficiency under different neighborhood backgrounds to investigate the impact of neighborhood background on forest carbon sequestration efficiency transfer. Table 5 illustrates the spatial Markov transition probability matrix of carbon sequestration efficiency in Chinese forestry under different spatial lag levels from 2003 to 2018. According to the calculation results, the following findings were obtained: (1) The transfer probabilities of various clubs vary with spatial lag level, but the principal diagonal probability values were greater than the non-diagonal probability values, indicating that the carbon sequestration efficiency of Chinese forestry exhibits a club convergence distribution regardless of spatial lag level, but the convergence characteristics differ. (2) At low degrees of spatial lag, the likelihood of stabilizing at both low and high efficiency was 100%, indicating that the transition from one state to another is deterministic. The transition probability matrix contains zero and one, indicating that every state in the Markov chain is stationary. As the probability of transitioning from one state to another was zero under low-level spatial lag, there is no periodicity and no cyclic transfer between states, and the system will not change the state at each time step. The long-term behavior is deterministic, and the system's state will not change regardless of how many time steps have passed. (3) At a low level of spatial lag, the probability of stabilizing at low efficiency was the highest (85.14%), greater than the 80.83% probability matrix of traditional Markov transfer. In contrast, the probability of maintaining relatively low, relatively high, and high efficiency was less than the probability of traditional Markov transfer. These results demonstrate that the convergence curing effect of all clubs (except low-efficiency clubs) was diminished, whereas the convergence effect of low-efficiency clubs was enhanced. In the transfer of different efficiency levels, adjacent clubs were more likely to transfer. (4) At a higher level of spatial lag, the club with higher efficiency had the highest probability (85.57%), followed by the club with lower efficiency (77.27%). Among them, only the club's lower efficiency convergence probability was smaller than that of the traditional Markov transfer probability matrix, indicating that the curing effect of club convergence exhibits an enhanced trend under a higher level of spatial lag. However, the convergence effect of "high-low" clustering was weakened. In the transfer of different efficiency levels, efficiency was mainly transferred upward. It can be seen that at a high level of spatial lag, forestry carbon sequestration efficiency presents a club convergence distribution. Nevertheless, the efficiency level of neighboring provinces will drive the efficiency improvement of surrounding provinces. (5) Under a high level of spatial lag, the probability of stabilizing in the high-efficiency club was the highest, reaching 100%, which is higher than the transfer probability under the traditional Markov chain, and the convergence probability of the other grade clubs was smaller than the traditional Markov chain. Compared with the transfer probability of other spatial lag conditions, the convergence effect of low-efficiency clubs was significantly weakened. Still, the convergence effect of high-efficiency clubs was considerably enhanced, indicating that the members of high-efficiency clubs continue to increase under a high level of spatial lag. Forming an excellent spatial layout of "high-high" clustering is easy. In short, efficiency is more

easily transmitted downward when there is less spatial lag, and efficiency is more quickly transferred upward when there is more spatial lag.

**Table 5.** The spatial Markov transition probability matrix of forestry efficiency in China from 2003 to 2018.

| Neighborhood Type | t\t + 1 | 1 | 2 | 3 | 4 | *n* |
|---|---|---|---|---|---|---|
| 1 | 1 | 1.0000 | 0.0000 | 0.0000 | 0.0000 | 3 |
| | 2 | 0.0000 | 0.0000 | 0.0000 | 0.0000 | 0 |
| | 3 | 0.0000 | 0.0000 | 0.0000 | 0.0000 | 0 |
| | 4 | 0.0000 | 0.0000 | 0.0000 | 1.0000 | 1 |
| 2 | 1 | 0.8514 | 0.1216 | 0.0135 | 0.0135 | 74 |
| | 2 | 0.2333 | 0.6333 | 0.1333 | 0.0000 | 30 |
| | 3 | 0.0000 | 0.0476 | 0.6667 | 0.2857 | 21 |
| | 4 | 0.0000 | 0.0000 | 0.2941 | 0.7059 | 17 |
| 3 | 1 | 0.7209 | 0.2791 | 0.0000 | 0.0000 | 43 |
| | 2 | 0.0341 | 0.7727 | 0.1818 | 0.0114 | 88 |
| | 3 | 0.0000 | 0.0879 | 0.7473 | 0.1648 | 91 |
| | 4 | 0.0000 | 0.0206 | 0.1237 | 0.8557 | 97 |
| 4 | 1 | 0.0000 | 0.0000 | 0.0000 | 0.0000 | 0 |
| | 2 | 0.0000 | 0.6500 | 0.3500 | 0.0000 | 4 |
| | 3 | 0.0000 | 0.0000 | 0.7257 | 0.2743 | 14 |
| | 4 | 0.0000 | 0.0000 | 0.0000 | 1.0000 | 36 |

*3.3. Prediction of Forest Carbon Sequestration Efficiency Trend in China*

The probability of Markov transfer limit distribution is the distribution of various types of transfers reaching an equilibrium state. The limit distribution of Markov and spatial Markov is calculated, that is, by calculating the n-step probability transfer matrix of the state type of forestry efficiency (the limit distribution of the state transition when n) after adding the spatial lag condition. By evaluating the limited distribution of state types of forestry carbon sequestration efficiency in provinces under each neighborhood background, the long-term evolution and growth trend of forestry carbon sequestration efficiency in China can be adequately predicted.

Table 6 shows the long-term evolution trend of China's forestry carbon sequestration efficiency. According to the Markov chain's limit distribution matrix, the chances of China's forestry carbon sequestration efficiency stabilizing at low, very low, comparatively high, and high efficiency are 8.13%, 18.38%, 33.79%, and 39.70%, respectively. This indicates that in the future, most provinces in China will concentrate on high-efficiency levels in forestry carbon sequestration, followed by relatively high efficiency, with the least on low- and relatively low-efficiency levels. Compared to the transition probability matrix under the starting conditions, the number of provinces and cities with low efficiency is dropping, while those with comparatively low, relatively high, and high efficiency are growing. This indicates that the overall forestry carbon sequestration efficiency in China will continue to rise, exhibiting a continuous development towards better efficiency and providing good conditions for progressing towards higher efficiency levels.

**Table 6.** Trend prediction of forest efficiency change in China from 2003 to 2018.

| | State Type | | 1 | 2 | 3 | 4 |
|---|---|---|---|---|---|---|
| Spatial lag is not considered | Initial distribution | | 0.5484 | 0.1613 | 0.1290 | 0.1613 |
| | Stationary distribution | | 0.0813 | 0.1838 | 0.3379 | 0.3970 |
| Spatial lag consideration | Stationary distribution | 1 | 1.0000 | 0.0000 | 0.0000 | 0.0000 |
| | | 2 | 0.1583 | 0.1008 | 0.3721 | 0.3688 |
| | | 3 | 0.0250 | 0.2048 | 0.3520 | 0.4181 |
| | | 4 | 0.0000 | 0.0000 | 0.0000 | 1.0000 |

The transition of forestry carbon sequestration efficiency under varied spatial lag circumstances varies significantly, according to the limit distribution matrix of the spatial Markov chain. Compared to the transition probability matrix under the initial conditions, it was discovered that under low-level spatial lag conditions, the probability of being at low efficiency could be 100%, implying that future forestry carbon sequestration efficiency in China will have a "single peak" distribution. Under relatively low spatial lag conditions, the probabilities of converging at relatively high efficiency were similar. In contrast, other efficiency types were less distributed, indicating that under relatively low spatial lag, China's future forestry carbon sequestration efficiency will evolve from a "single peak" to a "double peak" distribution.

Under relatively high spatial lag conditions, the probabilities of relatively high and high efficiency were still higher than low and relatively low efficiency, suggesting that under relatively high spatial lag, the future forestry carbon sequestration efficiency in China will move towards a high-efficiency club. Under high spatial lag conditions, the likelihood of converging at high efficiency was 100%, indicating an ideal club convergence effect of "those near vermilion get stained red" (a Chinese proverb suggesting one takes the conduct of one's company). It can be observed that under high spatial lag conditions, future forestry carbon sequestration efficiency in China will undergo a qualitative change from the relatively low-efficiency "single peak" distribution to the high-efficiency "single peak" distribution, significantly enhancing its efficiency level under the drive of neighboring high efficiency and expanding the scale effect of the "high-high" efficiency cluster.

In summary, the long-term evolution trend of forestry carbon sequestration efficiency in China is positive, according to current growth trends; the efficiency level will progressively increase over time, and the efficiency distribution indicates a direction of high concentration. The effect of varied neighborhood backgrounds on the evolution of forestry carbon sequestration efficiency in provinces and cities was found to be uneven, with provinces and cities with low-efficiency areas necessitating more improvements in efficiency. It presents the situation of a low-efficiency "single peak" distribution. In cities adjacent to regions with higher efficiency, the types of carbon sequestration efficiency clustered to four, showing a "single-peak" distribution of high efficiency, forming an ideal state of "near Zhu red", and the level of carbon sequestration efficiency will be improved.

## 4. Discussion

This paper aimed to measure the efficiency of forest carbon sequestration in various regions of China, analyze its temporal and spatial evolution characteristics, and predict its long-term trends. This paper analyzed the empirical results from three dimensions: temporal characteristics, spatiotemporal evolution, and trend prediction, and explored forestry management strategies that can effectively ameliorate carbon sequestration efficiency and promote regional coordinated development.

From the perspective of temporal characteristics, the research results revealed (Figure 1) that the carbon sequestration efficiency of forestry in China has been continuously improving, roughly the same as the research conclusion of Tian et al. [35]. In recent years, China has vigorously adjusted and improved its forestry policy. After forestry reform, China's forestry carbon sequestration output and management efficiency have been improved. The results also showed (Figures 2–4) that the spatial differences in forestry carbon sequestration efficiency in China's four forest regions gradually weakened, and regional forestry coordinated and balanced development began to be progressively realized.

From the perspective of temporal and spatial evolution, the research results revealed (Figure 7) that the distribution of forestry carbon sequestration efficiency among provinces in China is obviously unbalanced, the concentration degree of regions with higher carbon sequestration efficiency is increasing, and the efficiency gap is gradually decreasing. Forest resources are abundant in the northeast forest area, and the southern forest area and the carbon sequestration forest project are relatively mature, so the forest carbon sequestration efficiency exhibited a trend of continuous improvement, which is roughly the same as

the research results of Xue et al. [55]. Sichuan, Yunnan, and Tibet mainly represented the southwest forest area, which has always maintained a high-efficiency level above one. However, the carbon sequestration efficiency in the northern forest area was found to be relatively backward. As a result, the low forest coverage rate, insufficient forest resource endowment, and complex site conditions in the northern region hinder the high-quality development of forestry. Therefore, this state should focus on providing advanced experience to the northern forest region, promote the distribution and flow of capital, technology, and talents, and help the coordinated development of the forestry industry in the area.

From the perspective of trend prediction, the research results revealed (Table 6) that adjacency to low-efficiency provinces will inhibit improving efficiency levels. In contrast, adjacency to high-efficiency provinces will help improve forestry carbon sequestration efficiency. Therefore, future forestry carbon sequestration efficiency will appear in low-concentration distribution patterns. Thus, this state should give full play to the positive technology spillover effect of the areas with high carbon sequestration efficiency to improve the efficiency level of the neighboring areas. Provinces and regions should strengthen cooperation and exchanges, form a regional linkage mechanism, and realize complementary advantages.

Based on the above research results, this paper puts forward improvement measures and policy suggestions to ameliorate the carbon sequestration efficiency of China's forestry and promote the coordinated development of regional forestry from internal structure optimization, regional complementarity, and coordinated development according to local conditions. These recommendations can break through resource constraints internally, complement regional strengths externally, and finally help achieve carbon neutrality and high-quality forestry development. The relevant policy recommendations are as follows:

(1) Adapt to regional resource endowments and optimize the internal structure of forestry. Internal structure optimization primarily involves planting structure optimization, forest structure optimization, and forestry industry structure optimization. First, we should select trees with high carbon fixation that respond to the local climate, soil, and ecological environment, and encourage and nurture high-yield and high-quality forest kinds. Improvements in tree disease resistance, drought resistance, and cold resistance should also be prioritized to guarantee healthy tree development and excellent yield and quality. Second, during this stage, we should focus on "adjusting structure, improving quality, and increasing carbon sink", continue to strengthen young forest care and the transformation of low-yield and low-efficiency forests, gradually improve forest structure, constantly improve stand quality, and actively promote the transformation of China's seriously lagging forest management to further develop its forestry's carbon sequestration potential. Lastly, vigorous adjustments should be made to the forestry industry structure, promoting the development of green ecological industries. This will drive the integrated growth of forestry with other sectors, such as the development of under-forest economies and forest–tourism integration, enhancing the added value of the forestry industry.

(2) Focus on highlighting the advantages of forest areas to achieve regional complementary benefits. The efficiency level of forestry carbon sequestration in different provinces exerts mutual influence and interactions. Under high-value locations, the radiative impact of forest carbon sequestration should be highlighted, and forest carbon sinks should fully exploit their radiative effect and advanced expertise to effectively stimulate the development of forest carbon sinks in other places [55]. The positive technological spillover impact of the high-value aggregation area should be effectively exerted to make it flow to the low-value provinces to elevate the regional balance and lower the polarization connection. To improve the effectiveness of carbon sequestration, neighboring provinces should establish a comprehensive ecological forestry cooperation mechanism and policy linkage mechanism, strengthen forestry ecological

cooperation and exchanges, promote resource complementarity and complementary advantages, and create a positive feedback loop of win–win cooperation.

(3) Promoting the coordinated development of forest regions according to local conditions, the southwestern forest region is endowed with abundant forest resources, occupying a prominent position nationally in terms of forest area and accumulation. Nonetheless, a technological lag remains in forestry practices. It has been proposed that greater emphasis should be placed on importing expertise and technological advancements to catalyze high-quality development in the region's forestry sector through innovative approaches. The southern forest region exhibits suboptimal forest quality benefits and overarching ecological functionality, compounded via low forest land productivity and an irrational forest structure. Leveraging the south's inherent climatic, hydrological, and socioeconomic advantages is essential to enhance the forestry sector's quality and returns holistically. In contrast, the northern forest region encompasses limited resources, a challenging ecological landscape, and a nascent forestry industry. A comprehensive initiative involving sand prevention and control (the Three-North Shelter Forest) and sandstorm source management projects for Beijing and Tianjin have been advocated. Concurrently, the initiation of forestry circular economy pilots, bolstered human capital development, technological backing, and fiscal incentives are of paramount importance. The northeastern forest domain, characterized by its rich forest resources, can maintain its carbon sequestration capacity more effectively through the refined management of forest assets, optimization of tree growth patterns, scientifically grounded and standardized forest protection measures, and minimizing anthropogenic and natural adversities inflicted upon forest resources.

## 5. Conclusions

The efficiency of forest carbon sequestration in China was estimated in this study by developing the DEA-SBM super-efficiency model. The distribution and evolution model of carbon sequestration efficiency in forestry in China were summarized using the kernel density estimation method. On this basis, the spatial and temporal development trend and spatial distribution characteristics of forestry carbon sequestration efficiency in China were analyzed, and its long-term evolution trend was predicted using the traditional and spatial Markov chain transfer probability matrices. The research findings were summarized as follows: (1) Based on the line chart, kernel density estimation chart, and spatial distribution chart, the carbon sequestration efficiency of forestry in China exhibited a trend of fluctuating growth. However, it was still at a low level in most regions, with a significant potential for improvement in the future. Each province's forestry carbon sequestration efficiency distribution revealed an uneven pattern with a highly distributed value and a low-concentrated value. In terms of space, the spatial pattern of forestry carbon sequestration efficiency in China exhibited the characteristics of "three pillars", primarily displaying the pattern that Heilongjiang, Fujian, and Tibet are the high-efficiency gathering areas, driving the forest carbon sequestration efficiency gradually inward. (2) In terms of spatial evolution characteristics, the results of the traditional Markov probability transfer matrix showed that China's forestry industry's carbon sequestration efficiency is stable, and there is a "club convergence" phenomenon, making cross-stage transfer challenging to realize in the short term. Compared to the spatial Markov probability transfer matrix, the geographical pattern is crucial in the dynamic transfer of forestry carbon sequestration efficiency. When a province borders a province with high carbon sequestration efficiency, the likelihood of transferring to a high-efficiency state increases; in addition, when a province is adjacent to a province with low carbon sequestration efficiency, the likelihood of a shift to an inefficient state will increase. Therefore, in the spatial pattern, the situation of "high agglomeration, low agglomeration, high driving low, low inhibition high" is gradually formed. (3) From a long-term standpoint, being adjacent to different types of provinces will exert varying effects on the long-term trend of carbon sequestration efficiency. Being close to low-efficiency provinces would restrict efficiency growth, but being adjacent to

high-efficiency provinces will boost forestry carbon sequestration efficiency. Long-term growth prospects for China's forestry carbon sequestration efficiency are reasonably hopeful, with most provinces converting into a higher-level type and displaying a favorable trend towards high values.

**Author Contributions:** Conceptualization, F.Q. and C.W.; Data curation, W.S. and S.Z.; Formal analysis, L.Z.; Investigation, S.Z.; Methodology, S.Z.; Visualization, Y.A.; Writing—review and editing, S.Z. All authors have read and agreed to the published version of the manuscript.

**Funding:** This work is supported by the Natural Science Foundation of Gansu Province (grant no. 22JR5RA155), the Higher Education Innovation Fund Projects in Gansu Province (grant no. 2021B-087), and the Improving Young Teachers' Scientific Research Ability Project in Northwest Normal University (grant no. NWNU-SKON2021-22).

**Data Availability Statement:** The data presented in this study are available on request from the corresponding author.

**Conflicts of Interest:** The authors declare that they have no known competing financial interest or personal relationships that could have appeared to influence the work reported in this paper.

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
