# Peer review of "Temporal and Spatial Changes and Trend Predictions of Forest Carbon Sequestration Efficiency in China Based on the Carbon Neutrality Goal"

_forests, doi:10.3390/f14122387_

Round 1
Reviewer 1 Report
Comments and Suggestions for Authors
Dear Authors,
The subject of the manuscript entitled “Temporal and spatial evolution and trend prediction of forest carbon sequestration efficiency in China based on carbon neutrality goal” fit the profile of “Forests” journal. The study delivers interesting findings and can be a source of valuable information for the local as well as for the international readers. Therefore, I suggest to accept the article after minor revisions.
Abstract
Kindly check minor spelling mistake. The abstract needs a better composition of words. Moreover, I would suggest to mention some important values (Results) in this section.
Introduction
Introduction is well documented. However, research gaps at some places are not completely described in this section. The authors only reviewed research articles from china. I would suggest to add some material regarding the worldwide trend of carbon sequestration in forests to make it more interesting for the readers.
Material and methods
Materials and Method section is well explained.
Results
Results are well indicated. Pay special attention to grammar and spelling mistakes.
Discussion
This portion needs improvements. The authors must some lines which should be more focused for highlighting the important finding of this work and also compare the findings with previous studies for better looking discussion. Relevant tables and figures must be cited in discussion section also. This will facilitate the reader to follow the flow of the article.
References
Check and format the refrences according to the journal format
Conclusion
Minor language issues must be addressed to improve quality of the MS.
Comments on the Quality of English Language
Overall, the English language of the article is fine, some minor spelling mistakes and language issues must be addressed by the authors.
Reviewer 2 Report
Comments and Suggestions for Authors
Comments
Temporal and spatial evolution and trend prediction of forest carbon sequestration efficiency in China based on carbon neutrality goal
Abstract
· Page 1 line 23 Between provinces and regions, efficiency levels differ significantly, and the spatial paÄ´ern demonstrates the "three pillars" features – put these three pillars should be stated. Provide the years here on which the analysis was based on.
Introduction
· Page 2, line 49 – sentences do not flow in each other and should be revised. The entire paragraph should be improved.
· Page 3 line 122, additionally, we projected its long-term evolutionary trend - include the projection period
Data source and index system construction
· Page 3 line 133, for the data accessed, demonstrate its quality attributes and how inconsistencies were addressed.
· Page 5 line 192, the equations need to be improved – some characters are in Chinese language not English. Same case on page 7 line 241
· Model validation – how was it validated?
Results and discussion
· Page 7, line 253 - Define indicators of efficiency and effectiveness in the methodology – the confusion is a lot around these two terms
· There are two sections of discussion – either present results only or both
Comments on the Quality of English Language
It needs to be improved, same statements are in Chinese but the work is good
Reviewer 3 Report
Comments and Suggestions for Authors
The article is devoted to the important and actual issue of carbon sequestration by forests. The most important strength of the article is its interdisciplinary approach using methods of mathematical analysis borrowed from economics and management theory. Very interesting results were obtained based on the analysis of kernel density and Markov probabilities of changes in carbon absorption efficiency. However, the article suffers from a number of inaccuracies and questionable use of some terms.
1. Terms and concepts. Coordination of terms and concepts in an interdisciplinary study is the most important condition for its correct understanding. Since the journal "Forests" is primarily in the ecological niche, it is very desirable to explain to ecologists the concepts used from economics and management theory, as well as to use other terms in the interpretation familiar to ecologists.
- DEA-SBM models and “club convergence”. It will be very useful to explain to readers in a nutshell what is DEA-SBM models and the meaning of “club convergence” concept.
- I strongly advise not to use the terms “evolution”, “evolutionary” etc. From the point of view of biology and ecology, there is nothing about evolution in this article. These terms can be easily removed, and this will make the article more clear and better understood. In the text they are sometimes simply superfluous, in other cases it is better to replace them with "trends", "changes", "development", "patterns" or any other. See detailed comments.
- The use of the term "radiation" is also misleading. In natural sciences this term means energy in the form of waves or particles including radioactivity. Please find other words. See detailed comments.
- It is necessary to accurately distinguish between afforestation (afforestation of previously treeless areas) and reforestation (restoration of forest on land that had recently been covered with forest) and indicate whether you are talking about afforestation or reforestation. These approaches to forest planting are fundamentally different and may have opposite environmental results.
- It is necessary to accurately distinguish between carbon sink (sequestration) and storage. The effectiveness of carbon sequestration and carbon storage can vary in opposite directions. The used term "carbon sink storage" is inexplicable.
2. Methods.
- It is necessary to explain exactly how the authors obtained the carbon sequestration data (see detailed comments).
- Correct the description of the Markov chain method in the Methods section, namely, include explanation of what “1, 2, 3 and 4” mean.
3. Subtitle of section 4 is advisable to change, since things are discussed here were not discussed in the article. Section 4 discussed directions for future research and transformation of the forestry industry. The subtitle should reflect this.
4. Correct citation format and list of references
Detailed comments (see also uploaded PDF file)
Line 1. I strongly advise not to use the term “evolution” in this article, but replace it with "trends", "changes", "development", "patterns" or any other. See main text of comments.
Line 14. It might be better to name SBM type of model in full. Not all readers are experts in this modeling approach.
Lines 15-16. The term “evolution traits” here is very confusing for a person with a biological background. These words are very similar to “evolution of traits” - one of the key concepts in the theory of evolution, which, however, has nothing to do with the content of the article.
Line 24. Without explanation, the meaning of this phrase is unclear.
Line 25. Not everyone knows what is “club convergence”. It is advisable to either briefly explain this term or name this feature in words that everyone can understand.
Lines 63-65. In contrast to the recognized positive role of reforestation (restoration of forest on land that had recently been covered with forest), the role of afforestation (afforestation of previously treeless areas) is very controversial, both in relation to carbon storage and water regulation
Line 70. Here and further in the article it is necessary to accurately indicate whether we are talking about afforestation or reforestation. These approaches to forest planting are fundamentally different and may have opposite environmental results.
Line 96. What does TFP mean?
Line 117. For a better understanding of the article by readers, it is advisable either here or in the “Methods” section to briefly explain what is DEA-SBM type of model.
Line 147. Table 1. The information about thousands of yuan and persons seems unnecessary, since further in the article there are no specific values of these indices. In addition, you write that the approach used evaluates the effectiveness relatively. Therefore, the reader is not interested in what is the order of magnitude of the quantity of yuan or person in these units of measurement.
Table 1. Are these data presented in the accounting system? Further (lines 162-163) you write that they only need to be included in it
Line 162-163. Does this mean that these data currently are absent in the forest accounting system?
Lines 164-165. Does it mean that the authors assessed the carbon sink themselves and did not take these indicators directly from forest accounting? If the authors did the calculations themselves, it is necessary to explain what is the method of "mature stock expansion". What does "carbon sink storage" mean?
Lines 169-171. These calculations need to be clarified, since sync and storage are different things, and they may have different prices.
Line 176. This is a very dubious statement. When all the ecosystem services from forests that are important to human health and the agricultural and household economies are taken into account, their value may exceed the value of wood and carbon.
Line 179. Superiority over what? Seems like an unnecessary word.
Line 185-205. Not all readers are experts in this type of modeling. Please explain in a nutshell how this model uses the indicators from Table 1
Lines 227 and 240. Tables 2 and 3. Are these tables necessary if they only contain combinations of row and column numbers and no other information? For an example here you can refer to Markov tables from the Results section
Lines 223-240. I did not find in this section an explicit explanation of what “1, 2, 3 and 4” mean. It is explained only in section 3.2.1, but it should be here.
Line 262. It would be very interesting and useful if you could name what ecological scientific approaches are being used to improve carbon sequestration efficiency
Line 271-272. I advise clearly note here that it has increased
Lines 306-308. In section 2.2.2 you use the term "kernel density". It is better to maintain uniformity of terms in the article.
Line 322. Figures 2-5. The word “evolution” seems like a completely unnecessary word in the captions to Figures 2-5.
Line 342. In the article you estimate carbon sequestration. Sequestration and storage of carbon are different functions, the effectiveness of which can vary in opposite directions.
Line 344. "wave" seems better not to repeat word "peak".
Line 377. The entire section 3 discusses spatiotemporal patterns or trends in carbon sequestration efficiency, which you, inappropriately for biologists, call evolution. Thus, in this subheading I find it more useful to note that this section talking about the probabilities of changes in carbon sequestration efficiency
Line 381. "spatiotemporal patterns" seems better.
Lines 385-387. This should be noted in the Methods section 2.2.3
Lines 389 and 506. Tables 4 and 5. It would be helpful to highlight in any way the diagonals under discussion.
Line 398. “Club convergence” is an economic term which is rarely used in the natural sciences. Please explain to readers in a nutshell its meaning
Line 407. "Spatial distribution" or "spatial pattern" seems better.
Line 417. Here and below it is necessary to indicate clearly whether you are talking about afforestation or reforestation (see previous comments)
Line 437. The word "construction" is more suitable for technical structures (buildings, roads, etc.)
Line 441. Figure 7. It would be very useful to show the boundaries of the four regions discussed.
Line 458. Perhaps this is a proverb? But it is misleading, so then you are talking about THREE pillars.
Line 459. "Radiation sources" means sources of energy in the form of waves or particles including radioactivity. Please find other words
Line 519. Confusion about efficiency levels
Line 532. "efficiency" seems better.
Line 563. It is advisable to change the title of section 4, since things are discussed here that were not discussed in the article. Here you are talking about directions for future research and transformation of the forestry industry. The subtitle should reflect this.
Line 564-566. This needs to be rephrased, since the article itself does not consider either measures or even factors for high carbon sequestration efficiency.
Line 631. Section 3.1 talks about reducing fluctuations.
Line 636. "Radiation sources" means sources of energy in the form of waves or particles including radioactivity. Please find other words
Line 644. The word “conversely” is not suitable here, since the first part of the sentence does not contradict the second.
Line 647. "High radiation" means intensive radiation of energy in the form of waves or particles including cases of radioactivity. Please find other words.
Line 648. “Evolutionary” is superfluous word here.
Line 649. “long-term trend” seems better

Comments on the Quality of English Language
